

# Multilingual validation of the short form of the Unesp-Botucatu Feline Pain Scale (UFEPS-SF)

Stelio P.L. Luna[1], Pedro H.E. Trindade[1], Beatriz P. Monteiro[2],
Nadia Crosignani[3], Giorgia della Rocca[4], Helene L.M. Ruel[2], Kazuto Yamashita[5],
Peter Kronen[6], Chia Te Tseng[7], Lívia Teixeira[1] and Paulo V. Steagall[2,8,9]

[1] Department of Veterinary Surgery and Animal Reproduction, School of Veterinary Medicine and Animal Science, São Paulo State University (Unesp), Botucatu, São Paulo, Brazil

[2] Département de Sciences Cliniques, Faculté de Médecine Vétérinaire, Université de Montréal, Saint-Hyacinthe, Québec, Canada

[3] Department of Clinics and Veterinary Hospital, School of Veterinary, University of Republic, Montevideo, Uruguay

[4] Department of Veterinary Medicine, Research Center on Animal Pain, University of Perugia, Perugia, Italy

[5] Department of Companion Animal Clinical Sciences, School of Veterinary Medicine, Rakuno Gakuen University, Ebetsu, Japan

[6] University of Zurich, Zurich, Switzerland

[7] Crown Veterinary Specialists, Lebanon, NJ, United States of America

[8] Department of Surgical Specialties and Anesthesiology, Medical School, São Paulo State University, Botucatu, São Paulo, Brazil

[9] Department of Veterinary Clinical Sciences, Jockey Club School of Veterinary Medicine, City University of Hong Kong, Hong Kong, SAR, China

Corresponding author
Stelio P.L. Luna,
stelio.pacca@unesp.br

## ABSTRACT

**Background**. Pain is the leading cause of animal suffering, hence the importance of validated tools to ensure its appropriate evaluation and treatment. We aimed to test the psychometric properties of the short form of the Unesp-Botucatu Feline Pain Scale (UFEPS-SF) in eight languages.

**Methods**. The original scale was condensed from ten to four items. The content validation was performed by five specialists in veterinary anesthesia and analgesia. The English version of the scale was translated and back-translated into Chinese, French, German, Italian, Japanese, Portuguese and Spanish by fluent English and native speaker translators. Videos of the perioperative period of 30 cats submitted to ovariohysterectomy (preoperative, after surgery, after rescue analgesia and 24 h after surgery) were randomly evaluated twice (one-month interval) by one evaluator for each language unaware of the pain condition. After watching each video, the evaluators scored the unidimensional, UFEPS-SF and Glasgow composite multidimensional feline pain scales. Statistical analyses were carried out using R software for intra and interobserver reliability, principal component analysis, criteria concurrent and predictive validities, construct validity, item-total correlation, internal consistency, specificity, sensitivity, the definition of the intervention score for rescue analgesia and diagnostic uncertainty zone, according to the receiver operating characteristic (ROC) curve.

**Results**. UFEPS-SF intra- and inter-observer reliability were ≥0.92 and 0.84, respectively, for all observers. According to the principal component analysis, UFEPS-SF is a unidimensional scale. Concurrent criterion validity was confirmed by the high
correlation between UFEPS-SF and all other scales (≥0.9). The total score and all items of UFEPS-SF increased after surgery (pain), decreased to baseline after analgesia and were intermediate at 24 h after surgery (moderate pain), confirming responsiveness and construct validity. Item total correlation of each item (0.68–0.83) confirmed that the items contributed homogeneously to the total score. Internal consistency was excellent (≥0.9) for all items. Both specificity (baseline) and sensitivity (after surgery) based on the Youden index was 99% (97–100%). The suggestive cut-off score for the administration of analgesia according to the ROC curve was ≥4 out of 12. The diagnostic uncertainty zone ranged from 3 to 4. The area under the curve of 0.99 indicated excellent discriminatory capacity of UFEPS-SF.

**Conclusions**. The UFEPS-SF and its items, assessed by experienced evaluators, demonstrated very good repeatability and reproducibility, content, criterion and construct validities, item-total correlation, internal consistency, excellent sensitivity and specificity and a cut-off point indicating the need for rescue analgesia in Chinese, French, English, German, Italian, Japanese, Portuguese and Spanish.

# INTRODUCTION

Historically, cats have their pain underestimated, and therefore undertreated, when compared to their companion animal counterpart, the dogs (*Capner, Lascelles & Waterman-Pearson, 1999*; *Lorena et al., 2014*; *Steagall & Monteiro, 2019*). The prototype Unesp-Botucatu Feline Pain Scale (UFEPS) was refined in 2011 (*Brondani, Luna & Padovani, 2011*) and the full scale validated in 2013 (*Brondani et al., 2013b*). Afterward, the development of other feline specific pain assessment tools (*Calvo et al., 2014*; *Reid et al., 2017*; *Evangelista et al., 2019*; *Belli et al., 2021*) improved recognition of pain-related behavior and veterinary health care attitudes towards the provision of analgesia (*Simon et al., 2017*; *Steagall & Monteiro, 2019*;).

According to the most recent revised definition of the International Association for the Study of Pain (IASP), pain is 'an unpleasant sensory and emotional experience associated with, or resembling that associated with, actual or potential tissue damage' (*Raja et al., 2020*). Pain is a multidimensional experience defined not only by sensation and intensity, it includes qualitative and temporal attributes involving the affective-emotional and individual cognitive dimensions (*Steagall & Monteiro, 2019*; *Raja et al., 2020*). According to IASP 'verbal description is only one of several behaviors to express pain; inability to communicate does not negate the possibility that a human or a nonhuman animal experiences pain' (*Raja et al., 2020*, page 2). Because humans are still not able to understand animals' verbal expression, the recognition and the magnitude of animal pain rely on objective and/or subjective clinical pain assessment methods (*Gaynor & Muir III, 2009*; *Steagall & Monteiro, 2019*).

Objective methods are observer-independent and less prone to observer bias. They include nociceptive tests (*Dixon et al., 2007*), physiological parameters (*Brondani, Luna & Padovani, 2011*; *Brondani et al., 2013b*), and locomotor activity, like force plate, accelerometer, and gait analysis (*Moreau et al., 2014*; *Klinck et al., 2015*). Although they are important for certain types of studies, they cannot be used solely, have not been fully validated, and are generally non-specific. They may be invasive, require equipment, demand physical contact and the evaluators' presence. Their correlation with acute pain is questionable (*Steagall & Monteiro, 2019*; *Nicholls et al., 2021*). Endocrine changes require laboratory analysis, which is time-consuming, costly and not accessible in real-time (*Smith et al., 1996*; *Cambridge et al., 2000*). Previous use of drugs, interference of the emotional state, and low correlation with acute pain restrict the use of simple and clinically applicable physiological measurements like heart and respiratory rates, salivation and pupil diameter (*Höglund et al., 2018*; *Cambridge et al., 2000*). In cats, systolic arterial blood pressure seems to be the only good indicator of acute pain (*Smith et al., 1996*; *Brondani, Luna & Padovani, 2011*; *Brondani et al., 2013b*). These characteristics make these methods applicable only to specific experimental conditions but difficult to perform in clinical research.

Subjective pain assessment methods are usually based on pain-related behaviors, are minimally invasive and do not rely on equipment use. Some instruments allow remote evaluation and facilitate pain assessment in the research and clinical setting (*Steagall & Monteiro, 2019*). However, before clinical implementation, a pain assessment instrument should be valid and reliable. Validity is the tool's effectiveness to measure what has been proposed to and involves the three 'C's (criterion, content and construct validity) (*Streiner, Norman & Cairney, 2015*). Reliability demonstrates how much this measure is error-free by assessing its internal consistency, test/re-test stability, intra (repeatability) and inter-reliability (reproducibility) (*Jensen, 2003*). These attributes standardize the evaluations to guarantee reproducibility among scientific studies. In veterinary practice, a pain assessment instrument must indicate the analgesic intervention score to guide decision-making (*Brondani et al., 2013b*; *Reid et al., 2017*; *Steagall & Monteiro, 2019*).

Unidimensional (visual analogue, simple descriptive and numerical) scales are more straightforward when compared to composite scales. However, they are subjective, have limited reproducibility (*Holton et al., 1998*; *Martinez-Martin, 2010*; *Belli et al., 2021*), and may have ambiguous meaning (*Martinez-Martin, 2010*) since they do not encompass the multiple dimensions of pain (*Robertson, 2018*). On the other hand, composite and multidimensional scales have better consistency and accuracy than unidimensional ones because they evaluate different dimensions of pain (*Martinez-Martin, 2010*). In cats, four composite multi-items, weighed scales have undergone psychometric testing and some reported validity for acute pain assessment. They are chronologically the UFEPS (*Brondani et al., 2013b*), Glasgow Feline Composite Measure Pain Scale (CMPS-Feline) (*Calvo et al., 2014*; *Reid et al., 2017*), Feline Grimace Scale (FGS) (*Evangelista et al., 2019*), and the UFEPS-short form (UFEPS-SF) (*Belli et al., 2021*).

The UFEPS shows 'evidence of validity, reliability and sensitivity at the level of a randomized control trial' (*Merola & Mills, 2016*, page 60). It is available for online training at http://www.animalpain.org. However, the UFEPS is long and time-consuming, as it
includes three subscales and 10 evaluation items. Using it in its entirety requires blood pressure measurement, which is not always feasible and may disturb cats. Appetite is another difficult variable to assess in real-time or when cats are being fasted, for example. Because the UFEPS is a biological and statistical multidimensional instrument, these physiological measurements may be excluded since each dimension (subscale) may be assessed separately and have their independently calculated intervention analgesia score.

To encourage widespread use of pain scales, they should be short, simple and easy to score. Our recently developed UFEPS-SF, together with UFEPS, underwent clinical validation against a control group, clinical pain, soft tissue and orthopedic surgery, overcoming some of the previous limitations of UFEPS (*Belli et al., 2021*). However, only real-time pain assessment was performed, evaluators were not blinded to the pain condition, and intra-observer reliability was not calculated (*Belli et al., 2021*). Therefore, the UFEPS-SF requires further validation *via* video-scoring in a blinded manner to calculate intra- and inter-observer reliability and a more in-depth statistical validation using a larger number of observers.

Finally, a crucial gap that limits the use of an instrument is language and culture. Simple literal translations do not assure the same consistency and accuracy when the instrument is translated to different languages (*Guillemin, Bombardier & Beaton, 1993*; *Beaton et al., 2000*; *Sousa & Rojjanasrirat, 2011*; *Streiner, Norman & Cairney, 2015*). Thus, to ensure scientific rigor, the instrument must be validated in the language and culture of use (*Beaton et al., 2000*; *Sousa & Rojjanasrirat, 2011*; *Streiner, Norman & Cairney, 2015*) to assert the same semantics as the original scale (*Sperber, 2004*), like reported for McGill questionnaire in people (*Maiani & Sanavio, 1985*; *Boureau, Luu & Doubrère, 1992*; *Kim et al., 1995*; *Lázaro et al., 2001*; *Varoli & Pedrazzi, 2006*; *McGlone, Guay & Garcia, 2016*). In feline medicine, the UFEPS is the only instrument for assessing acute pain-related behaviors with reported validation in five languages: English (*Brondani et al., 2013b*), Portuguese (*Brondani et al., 2012*; *Brondani et al., 2013a*), Spanish (*Brondani et al., 2014*), French (*Steagall et al., 2017*) and Italian (*della Rocca et al., 2018*).

This study aims to test the psychometric properties of the UFEPS-SF in cats in eight languages, based on the evaluation of intra- and inter-rater reliability, responsiveness, content, construct and criterion validities, principal components analysis, item-total correlation, internal consistency, specificity, sensitivity, and score for indication of analgesic intervention.

# MATERIALS & METHODS

This study followed the Consensus Based Standards for the Selection of Health Measurement Instrument (COSMIN) checklist and terminology for assessing the methodological quality of studies (*Mokkink et al., 2010a*; *Mokkink et al., 2010b*) and the ARRIVE guidelines 2.0 (*Percie du Sert et al., 2020*).

## Ethics committee on animal use approval

This prospective, randomized and blinded study was approved by the Ethics Committee on the Use of Animals by the School of Veterinary Medicine and Animal Science, University of

São Paulo State (Unesp) (protocol number 180/2015). This study used videos recorded in the perioperative period of cats submitted to ovariohysterectomy in a previously published study after owner's written consent (previous protocol number 20/2008) (*Brondani et al., 2013b*).

## Video analysis

Videos corresponded to the perioperative period of 30 cats submitted to ovariohysterectomy (M1—before surgery, used as a negative control data; M2—30 to 60 min after surgery; M3—4 h after intervention analgesia with morphine 0.2 mg/kg IM, ketoprofen 2 mg/kg SC and dipyrone 25 mg/kg IV; and M4—24 h after surgery). The methodology for anesthesia, surgery and postoperative care was performed as described before (*Brondani et al., 2013b*). At all time-points, including before surgery, the surgical site was covered with a wound dressing to prevent identification of the time-point and avoid observer's bias.

One of the authors not involved in the subsequent video analysis (LT) condensed the original videos (*Brondani et al., 2013b*) to 3–4 min in length. Videos were made available to eight evaluators: Beatriz Monteiro (English), Chia (Joy) Tseng (Chinese), Giorgia della Rocca (Italian), Hélène Ruel (French), Kazuto Yamashita (Japanese), Nadia Crosignani (Spanish), Peter Kronen (German), and Stelio P L Luna (Portuguese). The videos were randomized (randomizer.org) and blindly evaluated for the first time regarding the order of the cats and perioperative time-points for each cat. After observing each video, the evaluator scored whether they would administer rescue analgesia according to their clinical experience. Next, the scales were scored in the following order: (1) numerical—NS (from 1–10; 1 corresponded to the animal without pain-related behaviors and 10 the maximum possible pain), (2) simple descriptive—SD (1—without pain-related behaviors, 2—mild pain, 3—moderate pain and 4—severe pain), (3) visual analogue—VAS (line from 0 to 10 cm, where 0 corresponded to the cat without the presence of pain-related behaviors and 10 the maximum possible pain) and (4) UFEPS-SF (*Belli et al., 2021*). After one month, the evaluators observed the same videos in a new randomized order. They carried out the same analyses with the inclusion of the CMPS-Feline (*Reid et al., 2017*). The evaluators had a period of one month to carry out the evaluations of each phase.

The UFEPS was developed and refined in previous studies (*Brondani, Luna & Padovani, 2011*; *Brondani et al., 2013b*). Each category score ranges from 0 (pain-free) to 3 (the most intense pain-related behavior). The maximum UFEPS score is 30 points. The UFEPS-SF was originated from the UFEPS and the number of items was reduced from 10 to 4 (Table 1). A detailed description of the items is available in the original manuscript (*Brondani et al., 2013a*).

## Content validation

A committee composed of five experienced veterinary anesthesiologists from multiple institutions, who did not take part in the subsequent validation of the scale, independently analyzed each scale item (describing both normal and pain-related behaviors) as irrelevant (1), little relevant (2), relevant (3) and highly relevant (4).

From the equation $CVR = \frac{ne - \frac{N}{2}}{N/2}$, where *ne* is the number of evaluators who consider the item relevant (scores 3 and 4) and N is the total number of evaluators, items with

**Table 1** Adaptation from UFEPS (*Brondani et al., 2013b*) to UFEPS-SF.

| Subscale (dimension) | UFEPS | UFEPS-SF |
|---|---|---|
| Psychomotor change | Posture | Posture |
| | Comfort | |
| | Activity | Comfort, activity and attitude |
| | Attitude | |
| Pain expression | Reaction to palpation of the abdomen/flank | Reaction to palpation of the painful area |
| | Reaction to palpation of the surgical wound | |
| | Miscellaneous behaviors | Miscellaneous behaviors |
| | Vocalization | Not included |
| Physiological variables | Arterial blood pressure | Not included |
| | Appetite | |

CVR values of 0.99 were approved (Table 2). After analyzing the instrument, the same committee should check whether the items were clear, easy to comprehend, or unclear. In the latter case, they had to indicate which items were unclear and provide suggestions to make the scale clearer and understandable (*Mokkink et al., 2010a*; *Mokkink et al., 2010b*).

## Translation, back translation and semantic equivalence

All translators were fluent in English, and both translators and evaluators were native speakers of the respective language, except BPM (English version). The original English scale was translated to each target language by two independent translators without interaction between raters. A third translator, fluent in both languages, synthesized the first two translations into one version to avoid incongruities of content and significance (in case of any incompatibilities, a consensus was attained among the translators and each language investigator). A fourth individual, unaware of the original scale, back-translated the synthesized version of the scale from each language to English. The synthesized and back-translated versions were compared and reviewed by each language investigator of the study who made minor amendments to maintain semantic equivalence. If necessary, the scale was adjusted by consensus among the individuals involved in the translation process to ensure conceptual, semantic, and content equivalence (*Sousa & Rojjanasrirat, 2011*).

## Statistical analysis

The criteria used to validate UFEPS-SF were defined as outcome measures (*Belli et al., 2021*). The sample size was based on previous reliability guidelines by accomplishing 30 heterogeneous samples (cats) and at least three evaluators (*Koo & Li, 2016*). The following statistical analyses were performed by PHET based on the assessment of all evaluators' two-phase results using the R software in the RStudio integrated development environment [Version 4.0.2 (2020–06–22)], (RStudio, Inc. Boston, MA, USA). Only the second evaluation phase data were used for CMPS-Feline analyses because this instrument was assessed only at this phase. For the UFEPS, only the results of SPLL from the previous publication were used (*Brondani et al., 2012*; *Brondani et al., 2013a*). Unless otherwise stated, all analyses were performed by grouping the data of all observers, all time-points and both phases.

**Table 2  Short-form of the Unesp-Botucatu Feline Pain Scale—UFEPS-SF (*Belli et al., 2021*) (http://www.animalpain.org).**

| ITEM | Description | Score |
|------|-------------|-------|
| | **Evaluate the cat's posture in the cage for 2 min.** | |
| | Natural, relaxed and/or moves normally | 0 |
| | Natural but tense, does not move or moves little or is reluctant to move | 1 |
| 1 | Hunched position and/or dorso-lateral recumbency | 2 |
| | Frequently changes position or restless | 3 |
| | | **Tick where applicable** |
| | The cat contracts and extends its pelvic limbs and/or contracts its abdominal muscles (flank) | |
| | The cats' eyes are partially closed (do not consider this item if present until 1 h after the end of anesthesia) | |
| | The cat licks and/or bites the painful site | |
| | The cat moves its tail strongly | |
| 2 | *All above behaviors are absent* | 0 |
| | *Presence of one of the above behaviors* | 1 |
| | *Presence of two of the above behaviors* | 2 |
| | *Presence of three or all the above behaviors* | 3 |
| | **Evaluation of comfort, activity and attitude after the cage is open and how attentive the cat is to the observer and/or surroundings** | |
| | Comfortable and attentive | 0 |
| | Quiet and slightly attentive | 1 |
| 3 | Quiet and not attentive. The cat may face the back of the cage | 2 |
| | Uncomfortable, restless, and slightly attentive or not attentive. The cat may face the back of the cage | 3 |
| | **Evaluation of the cat's reaction when touching, followed by pressuring around the painful site** | |
| | Does not react | 0 |
| | Does not react when the painful site is touched but does react when it is gently pressed | 1 |
| 4 | Reacts when the painful site is touched and when pressed | 2 |
| | Does not allow touch or palpation | 3 |

## Intra-observer (repeatability) and inter-observer (reproducibility) reliability

The level of agreement of each observer with him/herself (intra-rater reliability) was estimated by comparing the two phases of evaluation within one month. For inter-rater reliability, the level of agreement between the eight evaluators was estimated. For both reliability analyses, data from the total scores of the unidimensional scales, each item and total scores of UFEPS-SF were used, as well as the need for rescue analgesia. Only inter-rater reliability was calculated for the CMPS-Feline.

The weighted kappa coefficient (kw) was used, with the weighted disagreements according to their distance to the square of perfect agreement, to assess the agreement of the items on the UFEPS-SF, CMPS-Feline, numerical and simple descriptive scales and the need for rescue analgesia. The 95% confidence interval (CI) was estimated. For VAS, the intraclass correlation coefficient (ICC) two-way random effects model, type agreement multiple raters/measurements and its 95% CI were used (*Cohen, 1968*; *Landis & Koch, 1977*; *Schuster, 2004*). For the sum of UFEPS-SF and CMPS-Feline, the ICC two-way random effects model, type consistency multiple raters/measurements and their 95% CI

were used. The interpretation of kw and ICC was very good 0.81–1.0; good: 0.61–0.80; moderate: 0.41–0.60; reasonable: 0.21–0.4; and poor <0.2 (*Altman, 1991*).

## Distribution of scores

A frequency distribution graph was constructed for each UFEPS-SF item using descriptive analysis at each time-point and in all time-points grouped in order to assess the importance and representativeness of the item.

## Multiple association

Principal component analysis was carried out to define the number of dimensions or domains determined by different variables that establish the extent of UFEPS-SF. According to the Kaiser criterion (*Kaiser, 1958*), the representative dimensions of the components with eigenvalue >1 and variance >20 and each item of the UFEPS-SF with a loading value ≥0.50 or ≤−0.50 were selected.

## Criterion validity

For the concurrent criterion validation test, the UFEPS-SF was compared with a 'gold standard' instrument, considered the CMPS-Feline (phase 2 data) and UFEPS (by using as reference the data of SPLL analyzed previously) (*Brondani et al., 2013a*). Interpretation of Spearman's correlation coefficient was: <0.19: very weak; 0.2–0.39: weak; 0.4–0.59: moderate; 0.6–0.79: strong, 0.8–1: very strong (*Evans, 1996*).

Predictive criterion validity was assessed by the percentage of observers that would not provide rescue analgesia when cats were pain-free before surgery (scores below de Youden index) and would provide rescue analgesia when cats were possibly suffering postoperative pain, before analgesia (scores equal or above the Youden index—please see below).

## Responsiveness

The scores of each item and the total score of the UFEPS-SF, CMPS-Feline, unidimensional scales and the need for rescue analgesia over time were compared. Data distribution was evaluated by normal quantile–quantile plot and histograms. As data were not normally distributed, the generalized linear mixed model was used. Bonferroni's post hoc test was used for comparisons over time. Logistic regression analysis was performed, followed by Bonferroni's post hoc test for the dichotomous variable 'need for rescue analgesia'. The model of dependent variables (scales) showed Gaussian distribution according to the quantile–quantile figures and histograms, therefore mixed linear models were used, followed by the Bonferroni's post hoc test. Moments, evaluators, gender and phases were included as fixed effects, and individuals were considered as a random effect (*Silva et al., 2020*). For interpretation, it is expected that the differences in scores would be ordered as follows: immediate postoperative period >24 h postoperative >4 h after rescue analgesia ≥ preoperative period.

## Construct validity

The construct validity was determined by the three-hypothesis test method considering that (1) postoperative pain scores are higher than preoperative scores, (2) the scores should decrease after administering analgesics (3) and over time. Internal relationships were

assessed by internal consistency, item-total correlation and principal component analysis, and relationships to other instruments (UFEPS and CMPS-feline). Crosscultural validity, which is part of construct validity, was assessed by comparing the repeatability of each translated version of the scale with its original version (*Mokkink et al., 2010a*; *Mokkink et al., 2010b*).

### Item-total correlation
To analyze scale homogeneity, inflationary items and the relevance of each UFEPS-SF item, Spearman correlation compared each item with the sum of the scores of UFEPS-SF, excluding the assessed item. Values between 0.3 and 0.7 were accepted (*Streiner, Norman & Cairney, 2015*).

### Internal consistency
Cronbach's α coefficient estimated the consistency (interrelation) of scores for each UFEPS-SF item (*Crombach, 1951*). Interpretation: 0.60–0.64 minimally acceptable, 0.65–0.69 acceptable, 0.70–0.74 good, 0.75–0.80 very good and $>0.80$ excellent (*Jensen, 2003*; *Streiner, 2003*; *Streiner, Norman & Cairney, 2015*).

### Specificity and sensitivity
Perioperative scores of the UFEPS-SF were transformed into dichotomous scores ('0' corresponded to absence of pain expression behavior for a given item; $\geq$ '1'—presence of pain expression behavior) and applied to the respective equations. Specificity (baseline scores) = TN/(TN + FP); where TN = true negatives (scores representing painless behaviors '0' at the time cats were supposed to be pain-free, *i.e.*, before surgery) and FP = false positives (scores representing pain expression behaviors $\geq$ '1' at the time cats were supposed to be free of pain, *i.e.*, before surgery). Sensitivity (postoperative time-point before rescue analgesia) = TP/(TP + FN); where TP = true positives (scores representing pain expression behaviors $\geq$ '1' at the time cats should experience pain, *i.e.*, after surgery and before rescue analgesia) and FN = false negatives (scores representing the absence of pain expression behavior '0' at the time cats were expected to suffer pain, *i.e.*, after surgery and before rescue analgesia).

For the total score of the scales, the percentage of cats with scores $<4$ and $\geq 4$ for UFEPS-SF, $<7$ and $\geq7$ for UFEPS, $<5$ and $\geq5$ for CMPS-Feline (*Reid et al., 2017*), $<4$ and $\geq4$ for numeric rate, $<3$ and $\geq3$ for simple descriptive, and $<31$ and $\geq31$ for the visual analogue scales before surgery and after surgery (before rescue analgesia) were considered specificity and sensitivity, respectively. These cut-off points were calculated by the ROC curve described below. Interpretation: excellent 95–100%, good 85–94.9%, moderate 70–84.9%, not specific or sensitive $<70\%$ (*Streiner, Norman & Cairney, 2015*).

### Determination of the intervention score for rescue analgesia
The score indicative of the intervention point for rescue analgesia was calculated by the requirement for analgesia in the face of clinical experience (based on the first answer given by the evaluators after watching the videos) and considered the true value. Each item or the sum of the scales was the predictive value to build a ROC curve.

The calculation of the area under the curve (AUC) indicates the discriminatory capacity of the test. The ROC curve and the AUC is the graphic representation of the relationship between 'true positives' (sensitivity) and 'false positives' (1—specificity). The Youden index determined by the ROC curve is the simultaneous point of greatest sensitivity and specificity. The highest value of the Youden index = (Sensitivity + Specificity) − 1, represents the cut-off point for analgesic intervention. An AUC ≥0.95 indicates high discriminatory capacity of the scale (*Streiner & Cairney, 2007*).

The diagnostic uncertainty zone was determined by two methods, by calculating i. the Youden Index 95% confidence interval, replicating 1001 times the original ROC curve by the bootstrap method, and ii. the interval between the sensitivity and specificity values 0.90. The diagnostic uncertainty zone, which indicates the diagnostic accuracy, was the lowest and highest value of these two methods (*Cannesson et al., 2011*; *Celeita-Rodríguez et al., 2019*).

Another approach used to calculate the UFEPS-SF cut-off point was by using as the true value UFEPS scores ≥7 for sensitivity (true positives—time-point after surgery, before rescue analgesia) and <7 for specificity (true negatives—time-point before surgery).

The frequency and percentage of cats scored at baseline and after surgery (before rescue analgesia) in the diagnostic uncertainty zone of the cut-off point and the coherence for using rescue analgesia after surgery (before rescue analgesia), according to clinical experience and to the Youden index, were calculated by descriptive statistical analysis.

## RESULTS

The English and the translated versions of UFEPS-SF is presented in Table 2 and Table S1 (http://www.animalpain.org).

### Intra-observer (repeatability) and inter-observer (reproducibility) reliability

Repeatability was very good (>0.8) for all scales and items (Table S2). For all observers, UFEPS-SF ICC was ≥0.92 (minimum and maximum CI for all observers 0.89–1).

The inter-observer agreement (ICC; CI) of all evaluators was very good for UFEPS-SF (0.84–0.97; 0.80–0.98) and good for VAS (0.79–0.95; 0.77–0.96) and CPMS (0.78–0.96; 0.79–0.97). Weighed Kappa (CI) for the NRS was 0.80–0.96 (0.8–0.96) and for SDS was 0.77–0.96 (0.76–0.96) (Table S3) (*Altman, 1991*; *Streiner, Norman & Cairney, 2015*).

### Distribution of scores

For each item of UFEPS-SF, the score 0 prevailed before surgery (M1) and after rescue analgesia (M3). Scores 2 and 3 predominated at M2, a moment that scores were expected to be the highest after surgery, and scores 0, 1 and 2 were present at the moment of moderate pain (M4—24 h after the end of surgery) (Fig. 1).

### Multiple association

According to principal component analysis only one-dimension resulted in eigenvalue >1 (3.38) and variance >20 (84.34) (Table S4). Loading values were 0.95 for posture and miscellaneous behaviors, 0.94 for attitude and 0.83 for reaction to palpation. Therefore

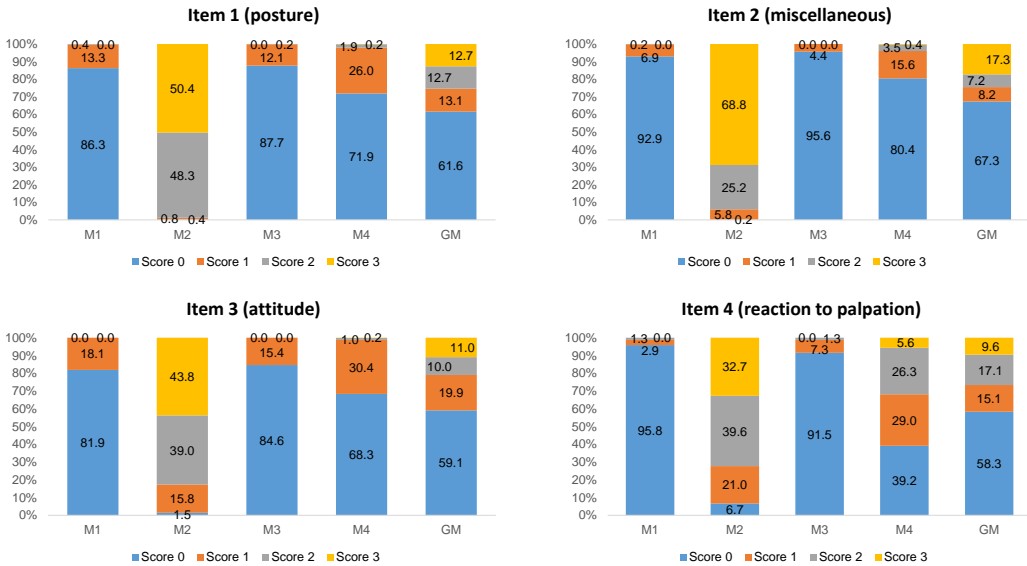

**Figure 1** **Frequency of occurrence of each UFEPS-SF item score.** M1—before surgery; M2—after surgery before rescue analgesia; M3—after surgery and rescue analgesia; M4—24 h after surgery. GM—data of all moments together (M1 + M2 + M3 + M4).

all items had loading values above 0.50 and were included in dimension 1 (Fig. 2). Only reaction to palpation had a loading value $<-0.5$ ($-0.56$) in dimension 2, however eigenvalue of dimension 2 was below the acceptable limit 1 (0.39). Therefore UFEPS-SF is unidimensional and confirmatory factor analysis seemed redundant.

## Criterion validity

Concurrent criterion validity was confirmed by the high correlation between UFEPS-SF with all other scales ($\geq 0.9$), specially CMPS-Feline and UFEPS, which may be considered the best representations of 'gold standard' instruments to assess pain-related behaviors in cats so far (Table S5).

## Responsiveness and construct validity

The scores of all scales, of all items of UFEPS-SF and of the need for rescue analgesia increased after surgery (pain) and decreased to baseline after rescue analgesia. Scores at 24 h after surgery (moderate pain) were intermediate between before and after surgery, showing that the scales were responsive to pain, to analgesic treatment and differentiated severe from moderate pain, therefore confirming responsiveness and construct validity hypothesis (Table 3; Figs. 3, 4 and 5). In all cases for each evaluator and gender, the UFEPS-SF results were the same as described above (Figs. 4 and 5). At all time-points, scores from female observers were higher than those of male individuals. The decrescent order of total scores were: German[1], Chinese[1,2], French/Italian/English[2,3], Spanish/Portuguese[3,4] and Japanese[4] (1 > 2 > 3 > 4). According to these results, there was an effect of gender and evaluator in the model (Figs. 4 and 5).

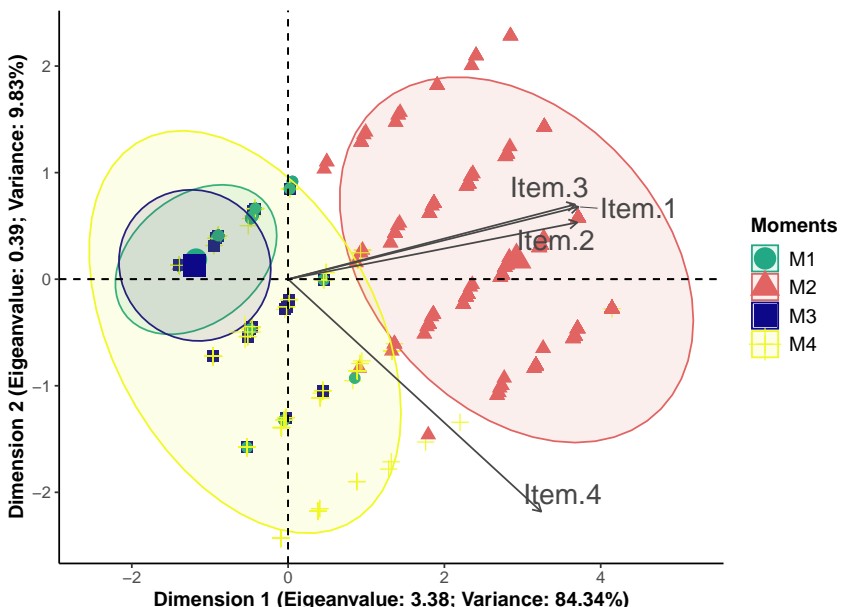

**Figure 2 Biplot of the principal component analysis of the UFEPS-SF.** UFEPS-SF—Unesp-Botucatu Feline Pain Scale–Short form. Confidence ellipses correspond to M1—before surgery (green); M2—after surgery, before rescue analgesia (red); M3—after surgery and rescue analgesia (blue); M4—24 h after surgery (yellow). The ellipse indicating the time when cats were in severe pain (M2) was positioned at the right side of the figure; on the opposite left side are the ellipses corresponding to the moments in which cats were probably not in pain (M1 and M3). The moment of moderate pain (M4) is positioned on both sides of the figure. All items on the scale are influenced by pain (M2) since their vectors are positioned in the direction of these ellipses.

**Table 3 Pain scores of the UFEPS-SF, CMPS-Feline, unidimensional pain scales and rescue analgesia indication before and after surgery, after rescue analgesia and 24 h after surgery in cats ($n = 30$).**

| Scales | Before surgery | | After surgery | | After rescue analgesia | | 24 h after surgery | |
|---|---|---|---|---|---|---|---|---|
| | **Median** | **Range** | **Median** | **Range** | **Median** | **Range** | **Median** | **Range** |
| Rescue analgesia | 0[c] | 0–0 | 1[a] | 0–1 | 0[c] | 0–1 | 0[b] | 0–1 |
| Numerical rating rate | 1[c] | 1–4 | 8[a] | 1–10 | 1[c] | 1–4 | 2[b] | 1–8 |
| Simple descriptive scale | 1[c] | 1–2 | 4[a] | 1–4 | 1[c] | 1–3 | 2[b] | 1–4 |
| Visual analog scale | 0[d] | 0–42 | 76[a] | 0–100 | 0[c] | 0–44 | 12[b] | 0–95 |
| Item 1 (posture) | 0[c] | 0–2 | 3[a] | 0–3 | 0[c] | 0–2 | 0[b] | 0–3 |
| Item 2 (miscellaneous) | 0[c] | 0–2 | 3[a] | 0–3 | 0[c] | 0–1 | 0[b] | 0–3 |
| Item 3 (attitude) | 0[c] | 0–1 | 2[a] | 0–3 | 0[c] | 0–1 | 0[b] | 0–3 |
| Item 4 (palpation) | 0[c] | 0–2 | 2[a] | 0–3 | 0[c] | 0–2 | 1[b] | 0–3 |
| UFEPS-SF | 0[c] | 0–5 | 9[a] | 0–12 | 0[c] | 0–4 | 2[b] | 0–12 |
| CMPS-Feline | 0[c] | 0–5 | 14[a] | 1–20 | 0[c] | 0–6 | 3.5[b] | 0–14 |

**Notes.**
UFEPS-SF, Unesp-Botucatu Feline Pain Scale—Short form; CMPS-Feline, Glasgow Composite Multidimensional Pain Scale (*Reid et al., 2017*).
Different letters express significant differences between moments where a > b > c > d, according to the mixed linear model (*Silva et al., 2020*).

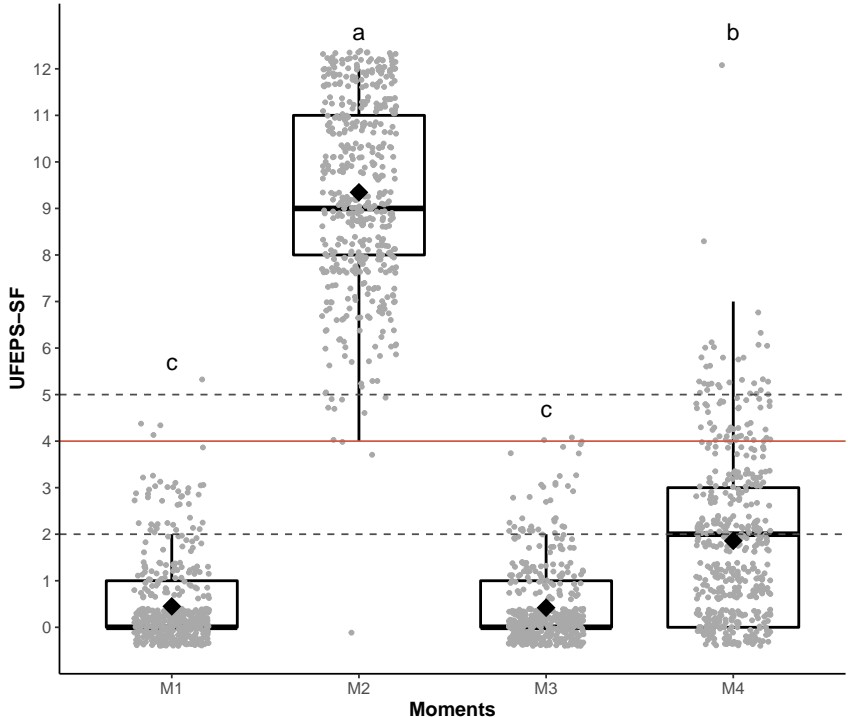

**Figure 3** **Boxplot of the perioperative UFEPS-SF total scores (median/amplitude) in cats submitted to ovariohysterectomy ($n = 30$).** The top and bottom box lines represent the interquartile range (25–75%), the line within the box represents the median, the extremes of the vertical lines represent the minimum and maximum values (mean $\pm$ 3 $\times$ standard deviation), black lozenges represent the mean, grey circles represent individual values and grey circles above or below the extremes of vertical lines represent outliers (above or below the mean $\pm$ 3 $\times$ standard deviation). UFEPS-SF—Unesp-Botucatu Feline Pain Scale–Short form. Different letters express significant differences between moments where $a > b > c$, according to the mixed linear model (*Silva et al., 2020*). M1—preoperative; M2—postoperative, before rescue analgesia; M3—postoperative, after rescue analgesia and M4—24 h postoperative.

## Item-total correlation

Item total correlation was above 0.3 when excluding each item separately, ensuring that each item contributed homogeneously to the total score (*Streiner, Norman & Cairney, 2015*) (Table 4). Except for reaction to palpation, the total score was minimally affected when each item was excluded. These items correlated well with the total score.

## Internal consistency

Internal consistency was excellent ($\geq$0.9) for all items. The very good interrelation of the items demonstrated the adequate internal structure of the scale (Table 4).

## Specificity and sensitivity

The specificity of the items ranged between 78 and 97%. Except for reaction to palpation that had good sensitivity (91%), sensitivity was excellent for the other items ($\geq$98%) (Table 5). Both specificity (calculated at baseline) and sensitivity (calculated after surgery) based on the Youden index were 99% (Confidence interval 97–100%).
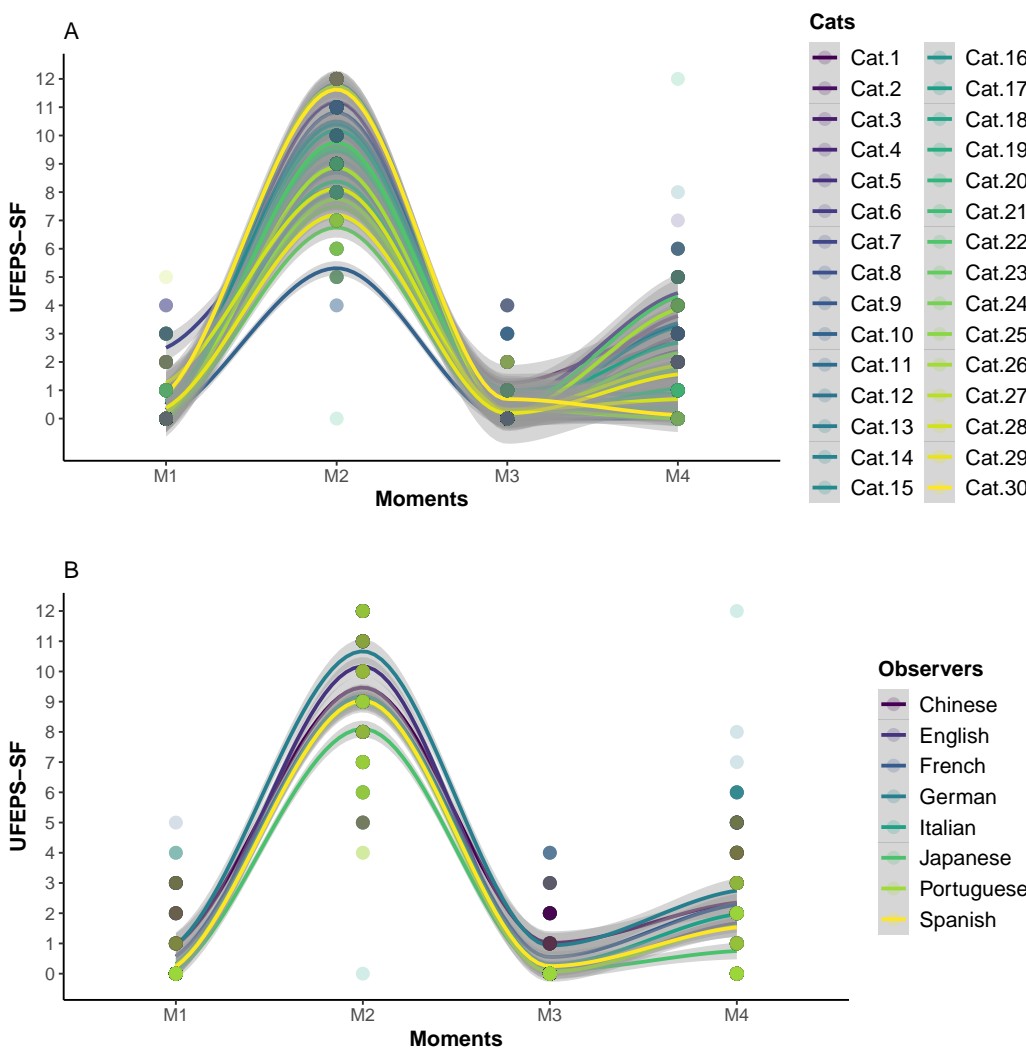

**Figure 4** Smoothed tendency lines, according to the LOESS method, indicating each cat (above) and each evaluator (below) total UFEPS-SF scores before (M1), after surgery, before rescue analgesia (M2), after rescue analgesia (M3) and 24 h after surgery (M4). The shaded area corresponds to the standard error of the smoothed lines. The colored circles represent the cats' (above) and the evaluators' (below) UFEPS-SF score distributions.

## Determination of the intervention point for rescue analgesia

The suggestive cut-off score suggesting the administration of analgesics according to the UFEPS-SF ROC curve was ≥4 out of 12 (Table 6; Fig. 6). Based on the two methods used to calculate the diagnostic uncertainty zone, the low and high confidence interval according to the bootstrap method was 3.5 and 3.5, respectively, and the interval between the sensitivity and specificity values 0.90 was between 2.5 and 4.7, therefore the last measure corresponding to the largest interval was used to define the diagnostic uncertainty zone for the UFEPS-SF and the other scales, as in most cases the last interval was the largest one (Fig. 6). According to this result, the diagnostic uncertainty zone scores ranged from 3 (≤2

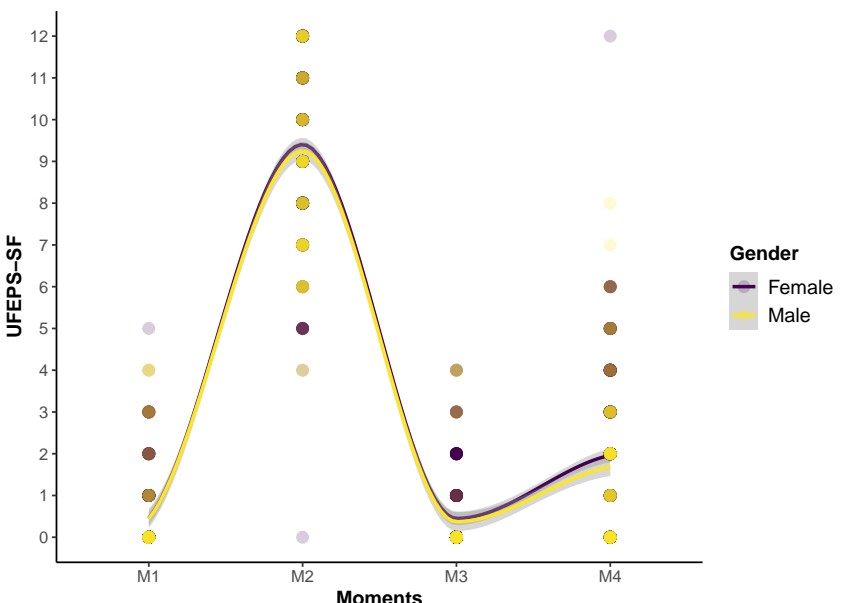

**Figure 5** Smoothed tendency lines, according to the LOESS method, indicating the male and female evaluators' total UFEPS-SF scores before (M1), after surgery, before rescue analgesia (M2), after rescue analgesia (M3) and 24 h after surgery (M4). The shaded area corresponds to the standard error of the smoothed lines. The colored circles represent the gender UFEPS-SF score distribution.

**Table 4** Item-total correlation and internal consistency of the UFEPS-SF.

| Items Tests | Item-total (Spearman) | Internal consistency (Cronbach's α) |
|---|---|---|
| Full scale | | 0.94 |
| | **Excluding each item below** | |
| Item 1 (posture) | 0.88 | |
| Excluding item 1 | 0.83 | 0.90 |
| Item 2 (miscellaneous) | 0.85 | |
| Excluding item 2 | 0.82 | 0.90 |
| Item 3 (attitude) | 0.88 | |
| Excluding item 3 | 0.81 | 0.91 |
| Item 4 (reaction to palpation) | 0.85 | |
| Excluding item 4 | 0.68 | 0.96 |

**Notes.**
UFEPS-SF, Unesp-Botucatu Feline Pain Scale—Short form.
Interpretation of Spearman's rank correlation coefficient ($r$): 0.3–0.7 (*Streiner, Norman & Cairney, 2015*). Interpretation of the Cronbach's α coefficient values: 0.60–0.64 minimally acceptable; 0.65–0.69 acceptable; 0.70–0.74 good; 0.75–0.80 very good; >0.80 excellent (*Streiner, 2003*).

indicates true negative pain-free cats) to 4 ($\geq 5$ indicates true positive—cats suffering pain). The same reasoning was adopted for the other scales. When palpation is not possible while using UFEPS-SF, the suggestive score for indication of rescue analgesia is $\geq 3$ (Table 6).

**Table 5   Specificity and sensitivity of the UFEPS-SF, CMPS-Feline and unidimensional scales.**

| Items Tests | Specificity (%) | | | Sensitivity (%) | | | AUC | Min | Max |
|---|---|---|---|---|---|---|---|---|---|
| | Estimate | CI | | Estimate | CI | | | IC | |
| | | Low | High | | Low | High | | Low | High |
| Item 1 (posture) | 86 | 83 | 89 | 100 | 99 | 100 | 91 | 93 | 94 |
| Item 2 (miscellaneous) | 93 | 90 | 95 | 100 | 99 | 100 | 95 | 96 | 98 |
| Item 3 (attitude) | 82 | 78 | 85 | 99 | 98 | 99 | 88 | 90 | 92 |
| Item 4 (palpation) | 96 | 94 | 97 | 93 | 91 | 95 | 93 | 95 | 96 |
| UFPES-SF (<4/≥4) | 99 | 97 | 100 | 100 | 99 | 100 | 99 | 99 | 100 |
| UFEPS (<7/≥7) | 100 | 88 | 100 | 100 | 88 | 100 | 100 | 99 | 100 |
| CPMS-Feline (<5/≥5) | 99 | 97 | 100 | 99 | 97 | 100 | 99 | 99 | 100 |
| Numerical rating scale (<4/≥4) | 98 | 97 | 99 | 98 | 97 | 99 | 100 | 99 | 100 |
| Simple descriptive scale (<3/≥3) | 100 | 99 | 100 | 97 | 95 | 98 | 99 | 99 | 99 |
| VAS (<31/≥31) | 99 | 97 | 99 | 97 | 95 | 98 | 99 | 99 | 100 |

**Notes.**
UFEPS-SF, Unesp-Botucatu Feline Pain Scale—Short form; UFEPS, Unesp-Botucatu Feline Pain Scale (*Brondani et al., 2013b*); CMPS-Feline, Glasgow Composite Multidimensional Pain Scale (*Reid et al., 2017*); CI, confidence interval.
Interpretation of specificity and sensitivity: excellent 95–100%; good 85–94.9%; moderate 70–84.9%; not specific or sensitive <70%; bold values ≥70% (*Bussières et al., 2008*).

**Table 6   Cut-off scores, specificity, sensitivity, Youden index and diagnostic uncertainty zone corresponding to intervention analgesia indication of the UFEPS-SF, UFEPS, CMPS-Feline and unidimensional scales.**

| Scale | Cut-off score | Specificity | Sensitivity | Youden index | Diagnostic uncertainty zone (scores) | |
|---|---|---|---|---|---|---|
| | | | | | True negatives (pain-free) | True positives (pain) |
| UFEPS-SF (0–12) based on indication of rescue analgesia | 4 | 96 | 96 | 0.91 | ≤2 | ≥5 |
| UFEPS-SF (0–12) based on UFEPS ≥7 | 4 | 96 | 96 | 0.95 | ≤2 | ≥5 |
| UFEPS-SF excluding palpation (0–9) | 3 | 95 | 92 | 0.87 | ≤1 | ≥4 |
| UFEPS (0–27) excluding blood pressure | 7 | 98 | 98 | 0.97 | ≤4 | ≥12 |
| UFEPS (0–24) excluding blood pressure and appetite | 7 | 98 | 97 | 0,95 | ≤4 | ≥11 |
| UFEPS pain expression only (0–12) | 2 | 84 | 98 | 0.83 | ≤2 | ≥3 |
| UFEPS psychomotor activity only (0–12) | 3 | 89 | 97 | 0.85 | ≤3 | ≥6 |
| CMPS-Feline (0–20) | 5 | 93 | 98 | 0.91 | ≤4 | ≥7 |
| NRS (1–10) | 4 | 97 | 97 | 0.94 | ≤2 | ≥5 |
| SDS (1–4) | 3 | 99 | 94 | 0.93 | ≤2 | ≥ 3 ≥ |
| VAS (0–100) | 31 | 96 | 95 | 0.92 | ≤27,5 | ≥34,5 |

**Notes.**
UFEPS-SF, Unesp-Botucatu Feline Pain Scale—Short form; UFEPS, Unesp-Botucatu Feline Pain Scale (*Brondani et al., 2013b*); CMPS-Feline, Glasgow Feline Composite Measure Pain Scale (*Reid et al., 2017*); NRS, numerical rating scale; SDS, simple descriptive scale; VAS, visual analog scale.

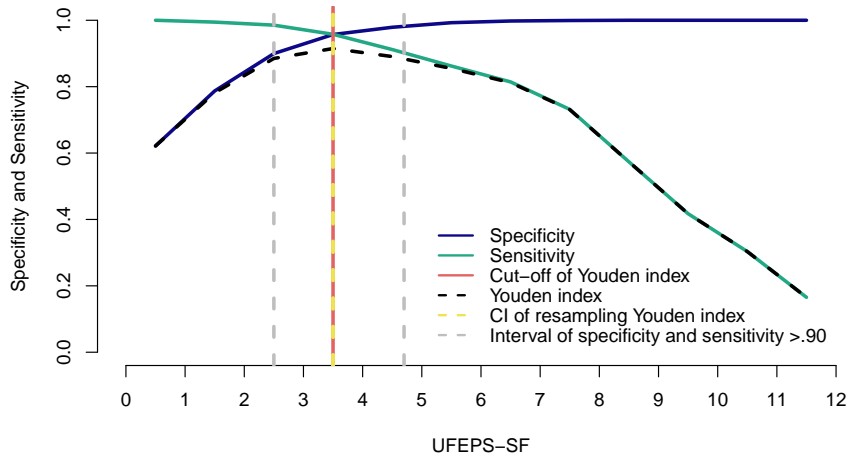

**Figure 6 Two-graph ROC curve with the diagnostic uncertainty zone for the UFEPS-SF.** Two-graph ROC curve, confidence interval (CI) with 1,001 replications, and sensitivity and specificity >0.90 applied to estimate de diagnostic uncertain zone of the cut-off point of all grouped evaluators, according to the Youden index for the Short-form of Unesp-Botucatu feline pain scale (UFEPS-SF) (*Cannesson et al., 2011*; *Celeita-Rodríguez et al., 2019*). The diagnostic uncertainty zone was from 3 to 4; <3 indicates pain-free cats (true negative) and >4 indicates cats suffering pain (true positives). The Youden index ≥4 represents the cut-off point for the indication of rescue analgesia.

The same cut-off point described above (≥4) was calculated for UFEPS-SF based on the UFEPS cut-off point ≥7 (excluding the physiological variables appetite and blood pressure) and so were the lower and upper scores of the diagnostic uncertainty zone (Table 6).

The cut-off score for the UFEPS excluding blood pressure is ≥7 out of 27, and by excluding the subscale physiological variables (blood pressure and appetite), it is ≥7 of 24. The cut-off point of the subscales pain expression and psychomotor activity were ≥2 and ≥3, respectively (Table 6) (*Brondani et al., 2013b*).

The AUC for all scales and subscales was above 97% (confidence interval = 95–100%). The AUC for the UFEP-SF was 99.2% indicating the high discriminatory capacity of all instruments (*Streiner & Cairney, 2007*). The percentage of cats showing scores inside the diagnostic uncertainty zone (scores 3 and 4) was low both when they were pain-free (≤5% for six evaluators and 16.7% for two evaluators) and when they were probably suffering the most intense pain (0% for six evaluators and 1.7% for two evaluators).

As presented before for sensitivity and specificity data, the percentage of cats where rescue analgesia was indicated before surgery, according to the Youden index (YI) of the UFEPS-SF (total score ≥4) was 0 for five evaluators, 1.7 for one evaluator and 3.3 for two evaluators. After surgery, rescue analgesia would be administered to all cats according to seven evaluators and to 98% of cats according to one evaluator. Therefore predictive criterion validity was confirmed for all evaluators as in most cases, cats would not be treated with rescue analgesia in the time-point they were pain-free (baseline) and would receive analgesia in the time-point they were possibly suffering pain after surgery.

## DISCUSSION

The UFEPS-SF is reliable and valid for acute pain-related behavior assessment in cats. Additionally, the cut-off for rescue analgesia guides the feline practitioner with the decision-making process on when to administer analgesics in the clinical setting. Both psychomotor change and pain expression subscales were summarized from four to two items, and physiological variables were removed. The original UFEPS was time-consuming and complex. These limitations have been addressed with the short-form making the instrument feasible to use and more practical while eliminating the need for appetite and blood pressure assessments (*Belli et al., 2021*). Appetite may not be easy to assess during fasting or postoperatively, for example, in cats undergoing gastrointestinal surgery or with a presence of an esophageal tube. Cats also tend to eat less on the first day of hospitalization (*Zeiler et al., 2014*). Blood pressure assessment is intrusive, demands equipment and may be inaccessible in feral or restless cats. However, because UFEPS is a multidimensional instrument, assessment of these items may be excluded, and the form can still be used since independent cut-off points are available for each subscale. Other limitations of both UFEPS and UFEPS-SF were addressed in the clinical validation study (*Belli et al., 2021*). In the latter, comparisons of these instruments with a negative control group and their use for assessing clinical pain and postoperative pain (including soft tissue and orthopedic surgery) were performed with reported validity. Additionally, the current study involved a large number of evaluators who were masked to the cat's clinical condition and pain state, the calculation of intra-observer reliability, and the assessment of validity and precision of the instrument in other languages (*Guillemin, Bombardier & Beaton, 1993*; *Beaton et al., 2000*; *Sperber, 2004*; *Streiner, Norman & Cairney, 2015*).

Validation was performed for each item separately, and rescue analgesia score was calculated without response to palpation, therefore, if necessary for convenience, the evaluator may choose to omit response to palpation of the affected area to avoid physical contact and/or disturbing the cat that may already be in pain, without compromising the psychometrical properties of the scale. To our knowledge, the UFEPS-SF is the only animal health care instrument available in the literature simultaneously assessing and validating its psychometrical properties in several languages and following the COSMIN checklist, taxonomy, terminology, and definitions (*Mokkink et al., 2010a*; *Mokkink et al., 2010b*). Previously, the UFEPS underwent cross-language validation in five languages (*Brondani et al., 2012*; *Brondani et al., 2013b*; *Brondani et al., 2013a*; *Brondani et al., 2014*; *Steagall et al., 2017*; *della Rocca et al., 2018*); however, this was performed with different studies rather than simultaneously, as presented herein. It is important to ensure that the results of studies performed in different geographic, cultural, language, and ethnic regions worldwide are reproducible and interchangeable. In the present study, the study design followed current guidelines for health care instrument translation with some adaptations (*Sousa & Rojjanasrirat, 2011*).

Content validation of the English version of UFEPS-SF was performed beforehand. All items were relevant and approved. Furthermore, the lowest confidence interval of

inter-rater reliability was above 80% for all language versions, so a pilot test for cognitive debriefing would not be necessary (*Sousa & Rojjanasrirat, 2011*).

Intra and inter-observer reliability of UFESP-SF were very good, ensuring that the instrument is respectively repeatable and reproducible like reported for UFEPS (*Brondani et al., 2012*; *Brondani et al., 2013a*; *Brondani et al., 2013b*; *Brondani et al., 2014*; *Steagall et al., 2017*; *della Rocca et al., 2018*). In this context, a minimum of 30 heterogeneous samples and three different evaluators should be used to assess the reliability (*Koo & Li, 2016*). In the current study, we used eight subjects and 120 videos (samples). Video-assessment results were slightly better than those obtained in the clinical study with both UFEPS and UFEPS-SF, where evaluators knew whether cats were possibly feeling pain because the assessment was performed in real-time and in-person, and evaluators were aware of the cats' pain state (*Belli et al., 2021*). In the current methodology, observer bias was avoided by randomizing videos on each phase (*Kaufman & Rosenthal, 2009*). However, the estimation of whether the cat required rescue analgesia before scoring any scale, may have still biased the evaluators to record higher scores to cats they judged to require analgesia or vice-versa. Evaluators did not receive any training but were individuals with experience in veterinary anesthesia and pain management. In rats, training improves the capacity to discern distinct degrees of pain regardless of observers' previous experience (*Roughan & Flecknell, 2003*). In terms of distribution of scores, they were proportionally increased according to pain intensity, confirming the importance and representativeness of each score level to reflect the degree of pain.

Principal component analysis provides an overview of the instrument by showing how the items associate with each other to determine how many dimensions compose the scale (*Gracely, 1992*). As expected for an instrument that measures pain, all items' eigenvectors were pointed to the moments of pain (after surgery and 24 h) and in the opposite direction to moments cats were supposedly pain-free (before surgery) or experiencing mild pain (after rescue analgesia). According to the Kaiser criterion, one component was selected, hence UFEPS-SF is unidimensional (*Streiner, Norman & Cairney, 2015*) as reported for other pain scales in cattle (*de Oliveira et al., 2014*), pigs (*Luna et al., 2020*), sheep (*Silva et al., 2020*), horses (*Taffarel et al., 2015*; *Barreto da Rocha et al., 2021*) and donkeys (*de Oliveira et al., 2021*). The only statistically defined multidimensional animal pain scale in cats is UFEPS (*Brondani et al., 2013b*); however, this classification is based on multivariate analysis and not on biological terms. Pain is a multidimensional phenomenon in nature and, in biological terms, components of both UFEPS-SF and CMPS-Feline represent not only pain intensity, but qualitative and temporal features, like sensory, motor, emotional, and cognitive dimensions (*Brondani et al., 2013b*). Except for the physiological variables, UFEPS-SF incorporates psychomotor (posture, comfort, activity and attitude), sensorial (reaction to palpation of the painful area), pain expression (miscellaneous behavior) and temporal features (response to analgesia); therefore, because UFEPS-SF covers many aspects of pain, it could be considered biologically multidimensional.

There are different methods to explore dimensionality and multiple associations between variables. In the present study, the structural models, observed according to principal component analysis, could have been verified by confirmatory factor analysis

by comparing a one-dimensional model (all items together in one dimension) *versus* a two-dimensional model (some items in one dimension and others in the second dimension) (*Bollen, 1989*). This is particularly relevant because the UFEPS-SF was derived from the UFEPS, a multidimensional scale. Usually, the selection of items for each dimension is based on loading values, which is the correlation between the item and the dimension. In our study, all four items showed loading values >0.80 in the first dimension and item 4 had a higher loading value in dimension 1 (0.83) than in dimension 2 (−0.53). Thus, there is no mathematical rationale to include item 4 alone in a second dimension and apply confirmatory factor analysis.

Criterion validity comprises concurrent and predictive validity. Concurrent criterion validity is the measurement of the strength of an instrument simultaneously compared to a previously validated instrument considered adequate to measure the target attribute. UFEPS was chosen as one of the 'gold standard' instruments since it has undergone a robust validity protocol (*Merola & Mills, 2016*). However, as UFEPS-SF was derived from UFEPS, the CPMS-Feline was also used for comparison (*Reid et al., 2017*), and so were the unidimensional scales, as previously reported in cats and other species (*Barreto da Rocha et al., 2021*; *Brondani et al., 2013b*; *Luna et al., 2020*; *de Oliveira et al., 2014*; *Taffarel et al., 2015*; *Silva et al., 2020*; *de Oliveira et al., 2021*). The correlations equal to or above 0.9 in all comparisons confirmed concurrent criterion validity.

Predictive criterion validation defines how predictable the practical results would be when using the instrument. In this case, the selected variable was the percentage of cats that should receive rescue analgesia, and therefore would have benefited, when they were suffering postoperative pain, and the percentage of cats that would receive unnecessary analgesia when they were pain-free before surgery. According to the sensitivity and specificity, 100% of cats would receive rescue analgesia when the highest pain scores were expected (true positives) and 1% of cats would be administered analgesics when they were supposedly pain-free (false positives), confirming adequate predictive criterion validity of UFEPS-SF. Even when considering the data of each evaluator and the Youden index, only one of the eight evaluators would not provide analgesia in 2% of the cats at the time-point they were possibly suffering pain, and one and two evaluators would provide analgesia in 1.7 and 3.3% of the cats when they were pain-free, respectively. Only a few cats had scores within the diagnostic uncertainty zone, safeguarding decision-making about whether rescue analgesia should be provided in the clinical setting.

All scales and all items of UFEPS-SF were responsive to pain and rescue analgesia and were capable of discriminating intense from moderate pain, which confirms responsiveness as reported with UFEPS (*Brondani et al., 2013b*) and its clinical validation (*Belli et al., 2021*). Construct validity was corroborated by the three-hypothesis testing: (1) pain-related behaviors increased after surgery and (2) decreased after analgesia and (3) changed over time as expected. These changes confirm that the UFEPS-SF measures the construct pain. Other COSMIN approaches to assess construct validity included: i. internal relationships, given by the results of internal consistency, item-total correlation and principal component analysis; ii. relationships to scores of other instruments (as reported for criterion validity); and iii. crosscultural validity (*Mokkink et al., 2010a*; *Mokkink et al., 2010b*). The assessment

of construct validity based on differences between relevant groups (*Mokkink et al., 2010a*; *Mokkink et al., 2010b*), has been performed in a previous publication where cats possibly experiencing pain in different clinical and surgical conditions showed higher UFEPS-SF pain scores than a negative control group of supposedly pain-free cats (*Belli et al., 2021*).

The exclusion of each item of UFEPS-SF influenced little and similarly item-total correlation, except reaction to palpation, suggesting that items have a first-hand association. The finding that the correlation was above 0.3 means that all items contribute to the scale. However, the correlation above 0.7 for its first three items may indicate that the scale is too specific, where one item may reiterate others (*Streiner, Norman & Cairney, 2015*).

Internal consistency was excellent for UFEPS-SF. When each item correlates well with the total score it is expected that internal consistency reduces by excluding the target item. This was the case for all items, except reaction to palpation, as mentioned for item-total correlation. According to the principal component analysis, this might be explained by the fact that reaction to palpation fits in two dimensions. Otherwise, the other items fit in only one dimension.

Any health instrument used for a specific diagnosis should correctly identify true positive (sensitivity) and negative individuals (specificity). The UFEPS-SF is specific and sensitive for acute pain-related behavior assessment in cats. The narrow diagnostic uncertain zone guarantees that cats suffering pain will likely receive analgesia and pain-free individuals will not be unnecessarily treated with analgesics. The high AUC of all scales and items of UFEPS confirmed the high discriminatory capacity of the tests and items.

This study presented some limitations. Only one observer for each language version was used, but because both intra and inter observer reliabilities were very good, their results were similar. Observers were experienced and, because in laboratory animals, training and experience enhanced pain recognition (*Roughan & Flecknell, 2006*), the next step is to validate the reproducibility of these results from naive or less experienced observers. Observations based on video analysis do not correspond to real-time assessment. In the current study, behaviors were condensed in short-edited videos, which might favor the observation of behavior exhibits and occurrences, especially of less frequent behaviors. The observer could pause and rewatch videos, possibly improving the detection of clinical signs of pain and reliability. A positive side of remote analysis is that real-time observation assessed by the in-person evaluator might affect pain expression as reported in rabbits (*Pinho et al., 2020*). However, Feline Grimace Scale scores were not different between real-time and video assessment (*Evangelista et al., 2020*) and were not affected by the presence of a caregiver during pain assessment (*Watanabe et al., 2020*). This limitation has already been addressed in the previous clinical validation study of UFEPS-SF, based on real-time comparisons (*Belli et al., 2021*). This previous study has shown that the UFEPS-SF may be used for different types of pain, including medical pain and postoperative assessment of soft tissue and orthopedic surgery (*Belli et al., 2021*).

Although the differences in time-point results were the same for evaluators, gender and when data were grouped, the influence of gender and evaluator is a potential bias in pain assessment. Three observers were male and five were female, which may be considered a small number of evaluators to assess gender effect. Women have more compassion towards

pain than men (*Sadeghiyeh, Khorrami & Hatami, 2012*; *Christov-Moore et al., 2014*). Female veterinarians attribute higher pain scores (*Williams, Lascelles & Robson, 2005*) and consider that dogs and cats experience more pain after surgery than male individuals (*Beswick et al., 2016*). Therefore based on the current study and the gender-related empathy towards pain, the effect of gender should be considered in the future when validating pain scales. In practical terms, the evaluator effect was not sufficient to modify the percentage of cats that would (after surgery) or would not (baseline) receive analgesia, according to the Youden index.

Although sample size was not calculated, the number of cats was based on the guidelines for reliability studies (*Koo & Li, 2016*) and on the minimum subject to item ratio of 5:1 for exploratory factor analysis (*Hair et al., 2014*). In the current study subject to item ratio was 7.5 (30 cats:4 items).

## CONCLUSIONS

The UFEPS-SF and its items, assessed by experienced evaluators, demonstrated very good intra-rater and inter-rater reliability, responsiveness, appropriate content, criterion and construct validity, item-total correlation, internal consistency, excellent sensitivity and specificity in Chinese, French, English, German, Italian, Japanese, Portuguese and Spanish languages. A cut-off score indicating the need to administer rescue analgesia was defined. Future studies are necessary to validate the reproducibility of these results when naive or less experienced observers assess the UFEPS-SF.

## ACKNOWLEDGEMENTS

The authors are grateful to Marilda O. Taffarel, Adriano B. Carregaro, Polly M. Taylor, Bradley T. Simon and Daniel Pang for content analysis and to the translators Grace Shih, Ya-Mei Cheng, Shelley Chi, Aurelien Ballaydier, Karina Klein, Anna Geks, Salim Darwiche, Alessandra Di Salvo, Maria Beatrice Conti, Kristinā Berardi, Ayako Oda, Sayaka Okushima, Yukari Miyake, Hiroki Sano, André Shih, Joao Henrique Neves Soares, Patricia de Queiroz-Williams, Karina Crosignani, Alicia Dib, Francisco Rosas and Angie Lagos.

### Funding

This work was supported by São Paulo Research Foundation (FAPESP) thematic research projects (2017/12815-0). The funders had no role in study design, data collection and analysis, decision to publish, or preparation of the manuscript.

### Grant Disclosures

The following grant information was disclosed by the authors:
São Paulo Research Foundation (FAPESP): 2017/12815-0.

### Competing Interests

The authors declare there are no competing interests.

## Author Contributions

- Stelio P.L. Luna conceived and designed the experiments, performed the experiments, analyzed the data, prepared figures and/or tables, authored or reviewed drafts of the paper, and approved the final draft.
- Pedro H.E. Trindade performed the experiments, analyzed the data, prepared figures and/or tables, authored or reviewed drafts of the paper, and approved the final draft.
- Beatriz P. Monteiro, Nadia Crosignani, Giorgia della Rocca, Helene L.M. Ruel, Kazuto Yamashita, Peter Kronen and Chia Te Tseng performed the experiments, authored or reviewed drafts of the paper, and approved the final draft.
- Lívia Teixeira performed the experiments, authored or reviewed drafts of the paper, contributed to video edition, and approved the final draft.
- Paulo V. Steagall conceived and designed the experiments, performed the experiments, authored or reviewed drafts of the paper, and approved the final draft.

## Animal Ethics

The following information was supplied relating to ethical approvals (*i.e.*, approving body and any reference numbers):

Ethics Committee on the Use of Animals (CEUA) of School of Veterinary Medicine and Animal Science (FMVZ)—São Paulo State University (Unesp)-Botucatu, Brazil approved this research (protocol number 180/2015).

## Data Availability

The raw data is available in the Supplemental File.

## Supplemental Information

Supplemental information for this article can be found online at http://dx.doi.org/10.7717/peerj.13134#supplemental-information.

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
