# Peer review of "Multilingual validation of the short form of the Unesp-Botucatu Feline Pain Scale (UFEPS-SF)"

_PeerJ, doi:10.7717/peerj.13134_

## Round 0.1 · original submission · Minor Revisions

Thank you for conducting this important work. I was fortunate to receive reviews from two expert pain researchers, both of whom recommended only minor revision. Both reviewers recommend reducing the number of tables and figures and placing some of the non-essential information into supplemental materials and I agree. The data in Tables 4 -6 could be incorporated into the text. Tables 11 and 12 probably are not needed. Consider combining Figures 4-6 as well. Consider removing Figure 8. Reviewer 1 points to the importance of defining pain clearly using the most recently and widely accepted definition. Reviewer 2 has some concerns about how validity has been impacted by the use of expert raters and asks you to provide clarifying information about whether they were naïve observers. I appreciate that the translation has been done rigorously using back translation. I have a few minor comments of my own. Line numbers refer to the reviewing PDF.
Lines 54-55, this is awkwardly worded “defined that they..??”
I don’t follow the point on lines 79-81. Be clear about how you are defining objective and subjective assessments and how these can be applied to nonverbal organisms.
Shouldn’t you have used CFA rather than PCA since you already had an idea of the dimensions you wanted the scale to assess?
Insert ; before however on line 451. Delete “only” and change “and not” to “rather than..”
Try to avoid single sentence paragraphs.
Line 551, this is not a “confounder.” Unless practitioners cannot rewind and rewatch videos in practice, why is this an issue? In any case, this would not constitute a confound.
I felt I needed a better sense of the nature of the scale’s items and how the questions related to the video before being presented with all of the validation statistics. In general, more structure would help reduce the feeling that the paper is just a laundry list of statistics.

Reviewer 1 ·

Basic reporting

The report by Luna and colleagues is clear and thorough and provides a wealth of references to the appropriate literature.

Experimental design

The methods are well-defined and rigorous. There is a lack of statistical details in the results and one is forced to read through the figure captions for this information. Some key details should be placed into the Results section. The breadth of the analyses is impressive, but most can be placed into the Supplemental section. Eight figures and twelve tables is too many for a journal article and dilutes the findings.

Validity of the findings

The findings are valid and can add to the assessment of pain-like behavior in cats across cultures and languages. The work can be quite impactful and has been rigorously tested.

Additional comments

1. There is an error in the Abstract: "...were randomly and evaluated.." seems to be missing a word.
2. The addition of the names and affiliation for the committee organized for content validation seems unnecessary and irrelevant. It can simply be stated that experts in the field from multiple institutions provided feedback.
3. In the Video Analysis section, lines 156 and 165 both state that the wound area was covered at all time points to avoid bias. Only one mention is needed.
4. As a more general comment, there is considerable disagreement as to the use of the term "pain". The most updated definition from IASP includes a note that verbal communication is not necessary, thus including non-human animals as experiencing pain. However, if in the M1 period, a cat is scored as showing pain when there is no reason to expect that pain is being experienced, it may be worth considering a statement that the scale assesses pain-related behaviors. In other non-human animal scales, it is understood that the closing of eyes or lack of body movement is a normal behavior that can be exacerbated by pain, but is not, in itself, a definitive sign of pain.

Reviewer 2 ·

Basic reporting

In this manuscript, Luna et al. present data demonstrating the validity of a recently developed feline acute pain scale in multiple languages. It represents a valuable addition to the literature as the recognition of feline pain is arguably substandard worldwide.
The article is well written and I have no real concerns in terms of the quality of the writing, the narrative or the article structure. I would urge the authors to consider reducing the numbers of figures and tables somewhat but including some the data as supplemental instead. The sheer number of figures and tables is overwhelming and perhaps distracting. I agree the data should be presented but most of the figures and some of the tables could be supplemental and simply referred to in the text.

Experimental design

The design of this study uses well accepted statistical techniques for validation of behavioural pain scales. The methods are well described and all relevant data is presented.
I would like to see clarification that all evaluators were viewing the videos for the first time as part of this study. If they had viewed these videos specifically beforehand, or perhaps used them extensively for education purposes then this would invalidate the results due to lack of blinding. Can this be clarified in the text?

Validity of the findings

The conclusions are supported by the findings and overall the scale performs admirably across different languages. Major potential issues with the study design are discussed well (e.g. the rationale for the use of video recordings)
The authors admit themselves in the discussion that they have only used 1 evaluator per language rather than the recommended 3. In addition all these evaluators are highly experienced. The validity of the scale may not be as good when used by a broader population of veterinarians who may well be less experienced and 'pain-savy'. While the authors already make this clear, I would like to see a mention of the validity being confined to experienced observers in the abstract and conclusions as I feel this represents an important limitation on the validity of the results.

---

## Round 0.2 · Minor Revisions

Thank you for responding to the reviewers’ requests. I still have some very minor issues for you to attend to before I can formally accept the paper.

It reads as if lines 80 and 81 should be combined.

Delete “thus” on line 85 as it is redundant with “Because…”

Please indicate page numbers for direct quotes (e.g., lines 128-129).

Line 146, change “does” to “do”

Line 563 and check throughout ; painful should be “in pain.”

I don’t understand what “Apart from it” refers to on line 572, place “a” in front of “few.”

Line 575, place “and” between “pain” and “rescue.”

Line 576, place “,” before “which.”

Line 609, I think there should be a , after animals.

Line 623, should “medical pain” be a procedure?

Please rephrase lines 630-631. What do you mean here? I am confused by whether there are effects of gender or not. If you found an effect of evaluator, you cannot say it is irrelevant unless you are saying that the specific pain level noted did not affect the recommendation of analgesia. Please be more explicit.

You discuss gender effects in two places in the discussion. Please eliminate redundancy and try to trim the discussion. It is still very long.
I don’t follow your logic on lines 637-639 – having reliability does not ensure having an adequate sample for statistical effects.

Table 3 is not necessary. Simply report the range of reliability in text.

---

## Round 0.3 · accepted · Accept

Thank you for making these minor corrections so promptly and thank you for performing this important work!